# Two Roles for the *Tenebrio molitor* Relish in the Regulation of Antimicrobial Peptides and Autophagy-Related Genes in Response to *Listeria monocytogenes*

**DOI:** 10.3390/insects11030188

**Published:** 2020-03-16

**Authors:** Maryam Keshavarz, Yong Hun Jo, Tariku Tesfaye Edosa, Yeon Soo Han

**Affiliations:** Department of Applied Biology, Institute of Environmentally-Friendly Agriculture (IEFA), College of Agriculture and Life Sciences, Chonnam National University, Gwangju 61186, Korea; mariakeshavarz1990@gmail.com (M.K.); bunchk.2000@gmail.com (T.T.E.)

**Keywords:** *Tenebrio molitor*, Relish, NF-κB, antimicrobial peptides, autophagy, *Listeria monocytogenes*

## Abstract

Relish is a key NF-κB transcription factor of the immune-deficiency (Imd) pathway that combats infection by regulating antimicrobial peptides (AMPs). Understanding of the fundamental role of *Tenebrio molitor* Relish (*Tm*Relish) in controlling of *Listeria monocytogenes* virulence through the regulation of both AMPs and autophagy-related (ATG) genes is unclear. Here, we show that *TmRelish* transcripts were highly abundant in the larval fat body and hemocytes compared to the gut upon *L. monocytogenes* infection. Furthermore, significant mortality was observed in *TmRelish*-silenced larvae after intracellular insult. To investigate the cause of this lethality, we measured the induction of AMPs and ATG genes in the *TmRelish* dsRNA-treated *T. molitor* larvae. The expression of *TmTenecin*-*1*, *TmTenecin*-*4*, *TmColeptericin*-*1*, *TmAttacin*-*2*, and *TmCecropin*-*2* were suppressed in the fat body and hemocytes of ds*TmRelish-*injected larvae during *L. monocytogenes* infection. In addition, *TmRelish* knockdown led to a noticeable downregulation of *TmATG1* (a serine-threonine protein kinase) in the fat body and hemocytes of young larvae 6 h post-infection (pi). The notable increase of autophagy genes in the early stage of infection (6 h pi), suggesting autophagy response is crucial for *Listeria* clearance. Taken together, these results suggest that *TmRelish* plays pivotal roles in not only regulation of AMP genes but also induction of autophagy genes in response to *L. monocytogenes* challenge in fat body and hemocytes of *T. molitor* larvae. Furthermore, negative regulation of several AMPs by *TmRelish* in the fat body, hemocytes, and gut leaves open the possibility of a crosstalk between Toll and Imd pathway.

## 1. Introduction

The innate immune system is an evolutionarily conserved defense mechanism that combats infectious pathogens. Once pathogens breach the first line of defense, the sensors, peptidoglycan-recognition proteins (PGRPs), recognize their respective bacterial peptidoglycans (PGNs) [1]. The signal generated from the PGRP/PGN interaction involves other intracellular components of the Toll and immune deficiency (Imd) signaling pathways, and it triggers nuclear translocation of nuclear factor kappa-light-chain-enhancer (NF-κB) transcription factors of those pathways, such as Dorsal and Relish, respectively [2,3]. Ultimately, this signal transduction leads to the transcriptional expression of immune-related effectors, such as antimicrobial peptides (AMPs) [4].

The main molecular components of Gram-negative bacteria and some Gram-positive bacteria (*Bacillus* and *Listeria* spp.) consist of *meso*-diaminopimelic acid DAP-type PGN, which binds to intracellular PGRP-LE and plasma membrane PGRP-LC, resulting in the activation of autophagy and the Imd pathway [5,6]. Moreover, PGRP-LE is not only capable of activation of Imd pathway, but also prophenoloxidase (proPO) cascade [7].

*Listeria monocytogenes*, a facultative intracellular gram-positive bacterium is able to invade and replicate in a broad range of host cells, including phagocytic and non-phagocytic cells [8]. Over several decades, this human pathogen became a host-pathogen interaction paradigm in vertebrates [9] and invertebrates [6,10]. Previous studies in *Drosophila* have demonstrated that PGRP-LE mediates the detection of *L. monocytogenes* invasion and is also responsible for initiating the autophagy pathway to suppress bacterial growth. Conversely, products of Relish, Dorsal, *Dif* (Dorsal-related immunity factor), and *Atg5* were not required for PGRP-LE-mediated suppression of *L. monocytogenes* growth [6].

Autophagy is a conserved process of self-dysfunctional cell degradation [11]. Autophagy genes are also incontrovertibly required for removing intracellular bacteria and parasites [12,13,14]. The autophagy-related (*ATG*) genes, which are involved in the formation of the autophagosome (Core autophagy machinery), consist of four main ATG protein complexes. The autophagosome initiation is firstly regulated by Unc-51 Like Autophagy Activating Kinase 1 (ULK1)/ATG1-ATG13 protein kinase complex, then Beclin 1 (BECN1)/ATG6-Phosphatidylinositol-4,5-Bisphosphate 3-Kinase Catalytic Subunit Epsilon (PIK3CE)/class III PI3K Vacuolar protein sorting 34 (Vps34) complex is responsible for autophagosome nucleation. Finally, two ubiquitin-like conjugation systems, which constitute ATG12-ATG5-ATG16L1 and ATG8-II, are required for expansion and completion [11]. Several autophagy-related genes have been identified in *Tenebrio molitor*, including *TmATG13* [15], *TmATG3* and -*5* [14], and *TmATG8* [13]. 

Recent investigations of *Drosophila* innate immunity have indicated that (stimulator of interferon genes (*dmSTING*) silencing, following infection with *L. monocytogenes,* leads to a reduction of immune-related gene expression and consequently, increased mortality. Consistent with these findings, *dm*STING has been shown to function through the activation of Relish (Imd pathway), leading to the induction of AMPs, to reduce *L. monocytogenes* load in *Drosophila* [16]. Relish is a fundamental downstream component of the *Drosophila* Imd pathway, however, previous studies have revealed that it plays an essential role in the onset of cell death and neurodegeneration via transcriptional upregulation of innate immune response genes, such as AMPs [17]. Moreover, it has been recently shown that Relish positively controls the programmed cell death and autophagy-related gene, *ATG1*, as a key activator of the autophagy pathway in the fly salivary gland [18]. 

The multifunctional pattern recognition molecule, PGRP-LE, acts upstream of Imd pathway and proPO cascade extracellularly while it activates Imd pathway and autophagy intracellularly. Consequently, it is reasonable to hypothesize that there might be a cross-talk between autophagy and Imd pathway [19]. Although it has been reported that *listeria*-induced autophagy is independent of Imd pathway in *Drosophila* [20], however, the association between autophagy and Relish-immune responses has been elucidated by earlier study in *Drosophila* models [18]. Similar results as reported for *Drosophila* PGRP-LE have been observed in the PGRP-LE homologue from the mealworm, *T. molitor* (Coleoptera: Tenebrionidae). *Tm*PGRP-LE controls the activation of autophagy, which is crucial for host survival against intracellular *L. monocytogenes* infection [10]. We wondered whether *TmRelish*, as a downstream transcription factor of PGRP-LE/Imd pathway, could directly or indirectly regulate both AMP genes expression and ATG genes level after *L. monocytogenes* challenge. To test our hypothesis, here, we utilized RNA interference (RNAi) methods. Our data present that *Tm*Relish is necessary for the regulation of AMP genes and autophagy-related genes in the fat body and hemocytes of *T. molitor* larvae.

## 2. Materials and Methods 

### 2.1. T. molitor Rearing 

*T. molitor* was maintained under standard conditions (at 27 ± 1 °C and 60 ± 5% RH) in the dark because they are nocturnal [21,22]. The insects were fed on an artificial diet consisting of 170 g of wheat flour, 0.5 g of chloramphenicol, 20 g of roasted soy flour, 0.5 g of sorbic acid, 0.5 mL of propionic acid, 10 g of soy protein, and 100 g of wheat bran in 200 mL of distilled water. The dietary preparation was autoclaved at 121 °C for 20 min and kept in an insectary. Only healthy 10^th^ to 12^th^ instar larvae (~2.4 cm) were used for all experiments.

### 2.2. Sample Collection after Immune Challenge with L. monocytogenes

*Listeria monocytogenes* strain American Type Culture Collection (ATCC) 7644 was grown in Brain-heart infusion (BHI) medium. In order to prepare the *L. monocytogenes* culture for microbial challenge studies, a single colony was chosen from *L. monocytogenes* BHI medium using a sterile wire loop and it was cultured overnight in 5 mL of fresh BHI broth at 37 °C at 200 revolutions per minute (rpm) in an orbital shaker. To prevent the death phase and to change the stationary phase to the exponential (log) phase, the overnight culture was subcultured in 5 mL of fresh BHI at 37 °C for 3 h. The new subcultured microorganism was washed three times with 1× PBS (phosphate-buffered saline, pH 7.0) at 3500 rpm for 5 min. The optical density of washed *L. monocytogenes* was measured at 600 nm and the microorganism was resuspended in PBS at a concentration of 1 × 10^6^ cells/μL.

1 μL of *L. monocytogenes* (1 × 10^6^ cells/μL) was injected into young *T. molitor* larvae in three independent experimental sets (n = 20). Challenged larvae were kept under insectary conditions and fed an artificial diet. Tissue samples including fat body, hemocytes, and gut were dissected from each infected individual and also from PBS-injected controls, at 3, 6, 9, 12, and 24 h post-injection (pi). First, a clean needle was inserted into the prothorax near the head of *T. molitor* larvae and pipette the larval hemolymph using a micropipette in 500 μL PBS buffer and centrifuged at 500 rpm at 4 °C for 15 min. The supernatant was discarded and the pellet (hemocyte) was washed with PBS buffer. To isolate the gut, cut the last abdominal segment and washed the dissected gut in PBS buffer using two forceps. To prevent the contamination of the fat body with Malpighian Tubules, the fat body was washed in the PBS buffer. Collected tissues were added to guanidine thiocyanate RNA lysis buffer (20 mM EDTA, 20 mM MES buffer, 3M guanidine thiocyanate, 200 mM sodium chloride, 40 μM phenol red, 0.05% Tween-80, 0.5% glacial acetic acid [for pH 5.5], and 1% isoamyl alcohol) and homogenized at 8500 rpm for 20 s using a homogenizer (Bertin Technologies, Bretonneux, France). The homogenized samples were then stored at −80 °C for total RNA extraction, cDNA synthesis, and quantitative reverse-transcription polymerase chain reaction (qRT-PCR).

### 2.3. RNA Extraction, cDNA Synthesis and qRT-PCR

Total RNAs were extracted from collected samples using the modified LogSpin RNA isolation method [23]. Homogenized samples were incubated at room temperature for 1 min and then centrifuged at 13,000 rpm at 4 °C for 5 min. Next, 100 μL of supernatant was transferred to 200 μL of RNA lysis buffer and the sample was then added to 300 μL of 99.9% ethanol. After gently inverting samples, they were transferred to a silica spin column (Bioneer, Daejeon, Korea; KA-0133-1) and centrifuged at 13,000 rpm for 30 s at 4 °C. The resulting aqueous phase was discarded. To digest genomic DNA, the silica spin column was treated with DNase (Promega, Madison, WI, USA; M6101) and incubated at 37 °C for 15 min. The silica spin column was then washed with 450 mL of 3 M sodium acetate buffer and then with 500 mL of 80 % ethanol and centrifuged as previously. After drying the spin column for 2 min, total RNA was eluted in 30 µL of distilled water.

For cDNA synthesis, 2 μg of total RNA was incubated with an oligo-(dT)_12–18_ primer for 5 min and then incubated in an AccuPower^®^ RT PreMix solution (Bioneer).

The prepared cDNA samples were diluted 1:20 prior to PCR analysis. To analyze the tissue distribution of *TmRelish*, qRT-PCR was performed using the cycling conditions recommended for the AccuPower^®^ 2X GreenStar qPCR Master Mix (Bioneer) (denaturation of 95 °C for 5 min, followed by 40 cycles of 95 °C for 15 s and 60 °C for 30 s) by adding *TmRelish*-specific primers. The *T. molitor ribosomal protein* gene (*TmL27a*) was used as a reference. Primers were designed using Primer3Plus (http://www.bioinformatics.nl/cgi-bin/primer3plus/primer3plus.cgi) and their sequences are listed in Table 1. qRT-PCR data were analyzed using the comparative C_T_ method (2^-ΔΔCT^ method; [24]. Statistical analysis of fold change values was performed using a Student’s t-test (*p* < 0.05).

### 2.4. Double-Stranded RNA TmRelish Synthesis

To prepare RNA interference (RNAi)-directed silencing of *TmRelish*, specific forward and reverse primers containing T7 promotor sequences at their 5′ ends were designed using SnapDragon-Long software (https://www.flyrnai.org/cgi-bin/RNAi_find_primers.pl). Amplicons of 851 bp were generated using AccuPower^®^ Pfu PCR PreMix, cDNA templates from *T. molitor* larvae, and specific primers (Table 1), under the following PCR conditions: 95 °C for 2 min, followed by 30 cycles of denaturation at 95 °C for 20 s, annealing at 56 °C for 30 s, and extension at 72 °C for 5 min. The generated products were purified using an AccuPrep^®^ PCR Purification Kit (Bioneer) and were then used to synthesize *TmRelish* RNAi in vitro with the EZ^TM^ T7 High Yield In Vitro Transcription Kit (Enzynomics, Daejeon, Korea), as per the manufacturer’s instructions. After incubating the mixture at 37 °C for 3 h and then 25 °C for 1 h, the synthesized dsRNA was gently mixed with one volume of 5 M ammonium acetate and incubated on ice for a further 15 min. The generated ds*TmRelish* was centrifuged at 13,000 rpm at 4 °C for 10 min and then washed three times with 70%, 80%, and 99.9% ethanol. After the final drying stage, the pellet was suspended in 30 μL of distilled water (Sigma, St Louis, MO, USA; W4502-1L). A dsRNA targeting enhanced green fluorescent protein (*EGFP*) was subsequently synthesized to serve as a negative control. 

### 2.5. Mortality Assay of TmRelish-Knockdown T. molitor Larvae Upon L. monocytogenes Infection

To understand the immunological role of *TmRelish* in response to *L. monocytogenes*, 60 healthy *T. molitor* larvae were divided into two groups. The first group (n = 30) was injected with double-stranded *EGFP* RNA, ds*EGFP*, as a negative control and the other group (n = 30) was injected with ds*TmRelish*. Three days after *TmRelish* RNAi injection, knockdown efficiency was verified (73%) and both groups were then challenged with *L. monocytogenes* (1 × 10^6^ cells/μL). Same sets (n = 60) were prepared for PBS-injected larvae as a control larval mortality was evaluated and *T. molitor* cadavers were removed from experimental plates daily for 10 days. The experiments were performed in triplicate.

### 2.6. Effect of TmRelish Gene Silencing on AMP Expression After Bacterial Insult

To assess the participation of *TmRelish* in humoral immunity and to determine whether high susceptibility of *TmRelish*-silenced larvae is related to AMP gene regulation, the mRNA expression levels of 14 identified AMPs, comprising *TmTenecin* family (*TmTenecin*-*1*, *TmTenecin*-*2*, *TmTenecin*-*3*, *TmTenecin*-*4*) [25,26,27,28], *TmAttacin* family (*TmAttacin*-*1a*, *TmAttacin*-*1b*, *TmAttacin*-*2*) [29], *TmDefensin*-*1*, *TmDefensin*-*2*, *TmColeptericin*-*1*, *TmColeptericin*-*2*, *TmCecropin*-*2*, *TmThaumatin like protein* family (*TmThaumatin like protein*-*1*, and *TmThaumatin like protein*-*2*) [30,31] were measured by qRT-PCR in fat body, hemocytes, and gut of ds*TmRelish-*injected larvae compared with ds*EGFP*-injected larvae after microbial infection. At 24 h post-*L. monocytogenes* infection, immune defense tissues were dissected in cold PBS solution, total RNAs were extracted and cDNA was synthesized as described above. AMP-specific primers (Table 1) were used for qRT-PCR.

### 2.7. TmRelish RNAi and Regulation of Autophagy-related Genes in Response to L. monocytogenes

To further understand the possible role of *TmRelish* in cellular immune responses, we assessed the expression levels of autophagy-related genes involved in the initiation, vesicle nucleation, and vesicle expansion and completion stages of autophagy. The relative mRNA levels of *TmATG1* (Atg9 cycling system), *TmVps34* which is a components of Phosphoinositide 3-kinase *(*PI3K) complex*, TmATG9* (Atg9 cycling system), *TmATG5* (Ubl system) [14], and *TmATG8* (Ubl system) [13] were analyzed using qRT-PCR in hemocytes, fat body, and gut of ds*TmRelish-*injected larvae compared with ds*EGFP*-injected larvae at 3, 6, and 9 h post-*L. monocytogenes* infection. The autophagy-related gene-specific primers were also designed using Primer3Plus and the primer sequences are shown in Table 2. PBS-injected and ds*EGFP*-treated larvae were used as the mock and negative controls, respectively.

### 2.8. Statistical Analysis


All experiments were carried out in triplicate and data were subjected to one-way analysis of variance (ANOVA). In order to evaluate the difference between groups (*p* < 0.05), Tukey’s multiple range tests were performed. The results for the mortality assay were analyzed using Kaplan–Meier plot (log-rank Chi-square test) by Excel (http://www.real-statistics.com/survival-analysis/kaplan-meier-procedure/real-statistics-kaplan-meier/).

## 3. Results

### 3.1. Expression and Induction of TmRelish Transcripts Upon L. monocytogenes Challenge

Previous studies of the immune function of *TmRelish* have shown that this transcription factor can be detected in all examined tissues, with high levels of expression in the gut, fat body, and hemocytes, and relatively low levels of expression in Malpighian tubules and integument [32]. Since the highest levels of *TmRelish* expression are in the most important immune-related tissues in *T. molitor*, these tissues were chosen to study the time-course of *TmRelish* expression using qRT-PCR. Following *L. monocytogenes* challenge, the mRNA levels of *TmRelish* in the fat body of challenged larvae were significantly upregulated from 3 h to 24 h, with the first peak at 3 h and the second peak at 9 h pi (Figure 1A). In hemocytes of *L. monocytogenes*-challenged larvae, the transcriptional levels of *TmRelish* reached their maximum at 6 h and then gradually decreased (Figure 1B). The expression levels of *TmRelish* in the gut of *T. molitor* larvae increased to 1.75-fold at 6 h, followed by 2.01-fold at 9 h, and then ultimately decreased to 0.5-fold at 24 h (Figure 1C).

### 3.2. Effect of TmRelish Gene Knockdown on T. molitor Larval Mortality after L. monocytogenes Infection

Given that *TmRelish* expression was modulated by *L. monocytogenes* infection, *TmRelish* is likely to play a key role in the immune responses against *L. monocytogenes*. Accordingly, *T. molitor* larvae may be more susceptible to *L. monocytogenes* after the RNAi-mediated depletion of *TmRelish*. To test this hypothesis regarding the functional role of *TmRelish*, survival was assessed in 10^th^–12^th^ instar *T. molitor* larvae injected with *TmRelish* dsRNA and then challenged with *L. monocytogenes* (1 × 10^6^ cells/insects). Larvae injected with dsRNA targeting *TmRelish* mRNA exhibited a knockdown efficiency of 73% at 3 days post-ds*TmRelish* treatment (Figure 2A). After 10 days, the final survival curve significantly differed in larvae injected with ds*EGFP* (90%) compared to those injected with ds*TmRelish* (57%) (Figure 2B). 

### 3.3. Effect of TmRelish RNAi on AMP Genes Expression in Response to L. monocytogenes

Our mortality data showed that *TmRelish* gene knockdown rendered *T. molitor* significantly more susceptible to *L. monocytogenes* infection, suggesting that *TmRelish* is a pivotal protein to combat infection with intracellular gram-positive bacteria. Next, to investigate the genes regulated by *TmRelish*, particularly those involved in the production of AMPs, we performed a *TmRelish* RNAi experiment and the transcript abundance of 14 AMP genes were subsequently measured by qRT-PCR. 

In the fat body of ds*EGFP-*injected *T. molitor* larvae, we observed clear upregulation of 10 AMPs: *TmTene1* and *TmTene4*; *TmDef1* and *TmDef2*; *TmCole1* and *TmCole2*; *TmAtt1a*, *TmAtt1b*, and *TmAtt2*; and *TmCec2* (Figure 3A). Meanwhile, the expression levels of these AMPs significantly decreased in the context of infection and *TmRelish* depletion. Conversely, the expression of antifungal AMPs such as *TmTLP1* and *TmTLP2* was notably higher in *TmRelish*-silenced larvae compared to ds*EGFP*-treated controls (Figure 3A). 

Furthermore, hemocytes of *TmRelish*-silenced *T. molitor* larvae displayed similar expression patterns for *TmTene1*, *TmTene4*, *TmAtt2*, *TmCole1*, and *TmCec2* (Figure 3B). Notably, seven AMPs such as *TmTene2*, *TmTene3*, *TmAtt1a*, *TmAtt1b*, *TmCole2*, *TmTLP1* and *TmTLP2* were not induced in the hemocytes of ds*EGFP-*injected groups while all aforementioned AMPs were barely detected in ds*TmRelish*-injected larvae (Figure 3B). Strikingly, *TmRelish* knockdown did not suppress the mRNA expression of *TmDef2* (Figure 3B). Collectively, the antimicrobial response of the larval fat body to *L. monocytogenes* infection was considerably higher compared to the larval hemocytes.

Interestingly, the mRNA levels of eleven AMPs including *TmTene1*, *TmTene2*, *TmTene4*, *TmDef1*, *TmDef2*, *TmAtt1a*, *TmAtt1b*, *TmAtt2*, *TmCole1*, *TmCole2,* and *TmCec2* showed a significant increase in the larval gut when *TmRelish* was silenced (Figure 3C). However, similar to hemocytes, there were no considerable changes in the transcriptional levels of antifungal AMPs namely *TmTene3*, *TmTLP1* and *TmTLP2* in the gut (Figure 3C). Taken together, our results suggest that *TmRelish* negatively regulates AMP expression in the gut of *T. molitor* whereas it acts as a positive regulator in the fat body in response to *L. monocytogenes* infection.

### 3.4. TmRelish Gene Contribution to the Regulation of Autophagy-Related Genes in Response to L. monocytogenes

The significant decrease in survival of ds*TmRelish* larvae upon *L. monocytogenes* infection led us to assess autophagy genes. To understand whether autophagy-related genes are affected when the transcription factor, *TmRelish*, is silenced, we performed RNAi experiments in the fat body, hemocytes, and gut of young larvae 3, 6, and 9 h pi.

In the fat body of *TmRelish*-knockdown larvae 3 h post-*L. monocytogenes* infection, *TmVps34* and *TmATG5* expression levels were considerably decreased compared with ds*EGFP*-treated controls. In contrast, the mRNA levels of *TmATG1*, *TmATG9*, and *TmATG8* were upregulated in *TmRelish*-silenced larvae (Figure 4A). In a similar pattern to what we found in the fat body of *TmRelish*-silenced larvae, *TmATG1* was induced at lower levels in the hemocytes and gut at 6 h pi (Figure 4). Silencing of *TmRelish* did not have an effect on *TmVps34, TmATG5,* and *TmATG8* expression in the fat body at 6 h pi (Figure 4A). However, intriguingly, the transcriptional levels of *TmATG1*, *TmVps34, TmATG9*, *TmATG5,* and *TmATG8* were markedly downregulated in the hemocytes of ds*TmRelish* larvae in comparison to ds*EGFP*-injected larvae at 6 h pi (Figure 4B). Additionally, in the hemocytes of *TmRelish*-silenced larvae, all examined autophagy-related gene expression levels were moderately upregulated, regardless of *TmATG9* at 3 h pi (Figure 4B). In contrast with hemocytes at 9 h pi, the expression levels of two autophagy-related genes comprising *TmATG1* and *TmVps34* were downregulated in the fat body of *TmRelish*-knockdown larvae (Figure 4A,B). While expression analysis showed a notable upregulation of *TmATG1, TmVps34,* and *TmATG5* expression in hemocytes of ds*TmRelish*-injected larvae at 9 h pi, also we observed no significant fold change in *TmATG9* and *TmATG8* expression in the same groups of larvae (Figure 4B). 

Similar to the fat body and hemocytes, *TmATG1* transcript was significantly increased in *TmRelish*-silenced groups at 3 h pi (Figure 4). Conversely, the transcript levels of *TmATG5* and *TmATG8* were slightly decreased in the gut of *TmRelish* dsRNA-treated *T. molitor* larvae at 3 h pi (Figure 4C). Of note, all the autophagy genes evaluated showed a significant upregulation in the larval gut against *L. monocytogenes* at 9 h pi (Figure 4C). 

Importantly, the mRNA levels of *TmATG1* were significantly decreased in all dissected tissues 6 h pi (Figure 4). These results suggested that *TmRelish* is crucial for the expression of *TmATG1* in the fat body, hemocytes, and gut 6 h pi.

## 4. Discussion

Insects are the most evolutionarily successful organisms and they can adapt well to the so-called pathosphere. The versatile immune system of insects provides them with a high capacity to eliminate pathogenic infections, which is a key factor in the successful dispersal of insects in the wild [33]. Among all insect models, *T. molitor*, a heat-tolerant beetle, has emerged as a useful host-pathogen interaction paradigm to investigate human pathogens, such as *L. monocytogenes* [34]. 

In recent years, it has been consistently reported that autophagy, as an innate immune response, plays important roles in the defense against intracellular infections [6,35]. Furthermore, the classical Imd pathway recognizes the DAP-type PGN, a major component of the cell walls of gram-negative bacteria and several gram-positive *Bacillus* and *Listeria* spp. [36]. Relish, an Imd transcription factor, engages a complex array of ancient defensive molecules (i.e., AMPs) to ward off microbial insults [37]. 

It is clear from the results presented here that *TmRelish* plays a significant role in the fat body, hemocytes, and gut of *T. molitor* larvae after *L. monocytogenes* challenge. *TmRelish* transcripts were highly abundant in fat body and hemocytes, suggesting that similar to other invertebrates, the immune response in *T. molitor* larvae also originates in both of these main immune-related organs [38,39,40].

Mortality assay was primarily performed to determine whether *L. monocytogenes* was capable of triggering a lethal infection in ds*TmRelish*-treated larvae. Our data showed that dsRNA knockdown of *TmRelish* resulted in an increased susceptibility of the larvae to *listeria*, suggesting the involvement of *TmRelish* in the defense against DAP-type bacteria. In congruence with our finding, the result of mortality assay in *TmPGRP-LE* RNAi-treated *T. molitor* larvae has revealed a significant death rate of insects in response to *L. monocytogenes* [10]. This was further supported by lethality of *Galleria mellonella* larvae after *L. monocytogenes* infection (1 × 10^6^ cells/μL) during 7 days [41]. Furthermore, a previous study in *Drosophila* reported a 100% mortality rate 6–8 days after *L. monocytogenes* infection [42]. Also, the implication of Relish (REL2) in mosquitoes immune responses (*Anopheles gambiae* and *Aedes aegypti*) has also been reported against other gram-positive bacteria such as *Staphylococcus aureus* [43,44]. Thus, understanding the reasons for the significant susceptibility of ds*TmRelish*-injected larvae to *L. monocytogenes* infection, is a major concern. Therefore, the abundance of 14 AMP genes transcripts was examined in *TmRelish*-depleted larvae after *L. monocytogenes* insult.

In *T. molitor* fat body, the expression levels of 10 AMPs were upregulated in non-*TmRelish*-silenced groups after *L. monocytogenes* infection; however, RNAi *TmRelish*-injected larvae showed downregulated expression of *TmTene1* and *-4*; *TmDef1* and *-2*; *TmCole1* and *-2*, *TmAtt1a*, *-1b*, and *-2*; and *TmCec2*. The results of the current study agree with the results of our previous studies on the gram-positive bacteria, *S. aureus*, in which all listed AMPs were found to be downregulated in the fat body of ds*TmRelish-*treated larvae following *S. aureus* challenge [32]. Therefore, *TmRelish* plays a critical role in response to Gram-positive bacteria, including intracellular bacteria in the larval fat body of *T. molitor*. However, *L. monocytogenes* similar to gram-negative bacteria contains DAP-type PGN, which is recognized by PGRP-LE in hemocytes of *Drosophila* [6]. As it has been confirmed, due to the strong elicitor activity of the DAP-type PGN, AMP expression levels of *Bombyx mori* are higher than lysine (Lys)-type PGN [45]. Furthermore, the antifungal AMP genes, *TmTLP1* and *-2* were highly expressed in the fat body of *TmRelish*-depleted larvae after infection with *S. aureus* and *L. monocytogenes*, suggesting that *TmRelish* negatively regulates antifungal AMPs. 

In the fat body, loss of *TmRelish* decreased expression of 10 AMP genes, noticeably weakening the *T. molitor* defense from infection with *L. monocytogenes.* Although we observed the increased levels of antifungal AMPs and *TmTene2*, suggesting *TmRelish* plays an important role in regulation of AMPs in the fat body. In the hemocytes of *TmRelish* knockdown larvae, five AMPs were similarly downregulated while eight AMPs were not affected by *TmRelish* dsRNA-injection. *TmRelish*, therefore, acts as a positive regulator for several AMPs in hemocytes. We also found that, in contrast to the fat body, in the gut of *TmRelish*-knockdown larvae, the expression levels of 13 AMPs were significantly increased by *L. monocytogenes*. Together, these data indicate that *TmRelish* is not required for the induction of AMPs in the gut of *T. molitor* larvae when infected with intracellular bacteria. This is in agreement with recent finding in *Manduca sexta* that *Ms*Rel2 and *Ms*Dorsal heterodimers suppressed AMPs activation [46].

AMP genes induction in hemocytes of ds*EGFP*-injected larvae after *L. monocytogenes* infection was comparatively lower than AMP genes induction in the fat body. Moreover, *TmTene1* and *-4*, *TmCole1*, *TmAtt2*, and *TmCec2* induction was at a noticeably lower level in hemocytes of *TmRelish*-silenced larvae, while the expression of *TmDef2* was increased. Our study revealed that *TmDef2* expression was significantly high in the hemocytes and gut of *TmRelish* knockdown larvae while the mRNA expression of *TmDef1* was strikingly strong in the gut. Early studies showed that *L. monocytogenes* was strongly inhibited by defensins [47,48]. Therefore, we propose that *T. molitor Defensin* family might function as a prominent antimicrobial peptide in the hemocytes and gut in response to *L. monocytogenes*. The presence of AMPs in the larval hemolymph of *Galleria mellonella* can inhibit the *L. monocytogenes* growth [41,49]. Similar to the results seen after *L. monocytogenes* infection, when *TmRelish* was knocked down, the expression levels of same AMPs were affected during *S. aureus* challenge [32]. Therefore, *Tm*Relish positively regulates five AMPs in *T. molitor* hemocytes in response to a Gram-positive bacterial infection. Consequently, these findings indicate that the expression of *TmTene1* and *-4*, *TmCole1*, *TmAtt2*, and *TmCec2* rely on *TmRelish* in the fat body and hemocytes of *T. molitor* larvae in response to a challenge by Gram-positive bacteria (*S. aureus* and *L. monocytogenes* (Figure 5).

In addition, it has been shown that Relish can cause autophagy (or cell death) with various effects, including the death of *Drosophila* photoreceptor cells in *norpA* mutant flies [50], the elimination of loser cells in *Drosophila* wing discs during cell competition [51], and *Drosophila* neurodegeneration [52]. Intriguingly, a dual role of Relish has recently been described, whereby it positively regulates the induction of *Atg1*, leading to the activation of autophagy in the developmental stage of *Drosophila* salivary glands and also regulates immune-related genes after microbial insult [16,18]. 

In hemocytes 6 h pi, a significant downregulation of autophagy genes was observed in *TmRelish*-depleted larvae, whereas this downregulation was only seen for *TmATG1* in the fat body at the same time point. In fact, our results showed that *TmRelish* regulated the expression of *TmATG1*, a core component of the autophagy process in both fat body and hemocytes in response to intracellular infection. Previous studies have demonstrated that *Drosophila ATG1* is critical for autophagy in the fat body and its expression is sufficient to affect the overall autophagy mechanisms [53]. Furthermore, the expression levels of *TmATG1* was noticeably downregulated in the fat body 9 h pi, but not in hemocytes and gut of *TmRelish*-silenced larvae. The relatively high expression of *TmATG1* in the fat body compared to other autophagy genes indicates the importance of this gene in mounting an autophagic response which is consistent with the previous reports [54,55]. Silencing of *TmATG8*, as an approved marker of autophagy, impaired the autophagic signal and reduced the resistance of *T. molitor* larvae against *Listeria* infection in hemocytes [13]. Similarly, we observed that *TmATG8* transcripts of ds*TmRelish* knockdown larvae slightly decreased in the hemocytes 6 h pi. These data provide compelling evidence that the reduced survival of larvae might be also related to autophagy dysfunction. Moreover, an increase expression of autophagy genes over time with the highest level 6 h pi in the ds*EGFP*-treated larvae after infection, suggesting that autophagy plays a major role in clearance of *L. monocytogenes* in the hemocytes. As has been previously reported, exposure of J774 cells to cytosolic *L. monocytogenes* lead to autophagosome formation 6 h after insult [56]. Furthermore, the rapid autophagic responses in the early phase of infection limited growth of *L. monocytogenes* following vacuole formation [57]. However, *L. monocytogenes* is able to escape auotophagic recognition via several surface proteins [58]. Regarding the PGRP family (PGRP-LE and PGRP-SA) and how it recruits both Imd and Toll pathway components to erase the invading and escaped *Listeria*, as well as autophagy [9,10,16]. One intriguing possibility is that *L. monocytogenes* is recognized by *TmPGRP-LE*. This recognition subsequently leads to *TmRelish* translocation which modulates not only AMPs induction via Imd signaling but also ATG genes expression in autophagy.

## 5. Conclusions

Our report demonstrates the immunological and autophagic function of *TmRelish* in various immune-related tissues of *T. molitor* during *Listeria* infection, through the regulation of AMP and ATG genes. The current investigation delineates the possible connection between different immune response mechanisms, including the Imd signaling pathway and autophagy.

## Figures and Tables

**Figure 1 insects-11-00188-f001:**
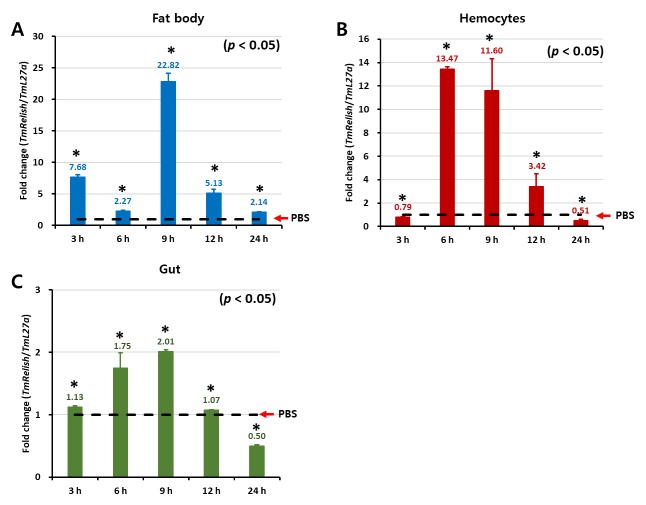
*TmRelish* mRNA expression levels in the fat body (**A**), hemocytes (**B**), and gut (**C**) upon *L. monocytogenes* infection. Total RNAs were isolated from young larvae (10^th^–12^th^ instar larvae) at 3, 6, 9, 12, and 24 h post-injection. *T. molitor 60S ribosomal protein L27a* (*TmL27a*) was used as an endogenous control. *TmRelish* expression in PBS-treated larvae was normalized to 1. ‘*’ shows significant differences (*p* < 0.05).

**Figure 2 insects-11-00188-f002:**
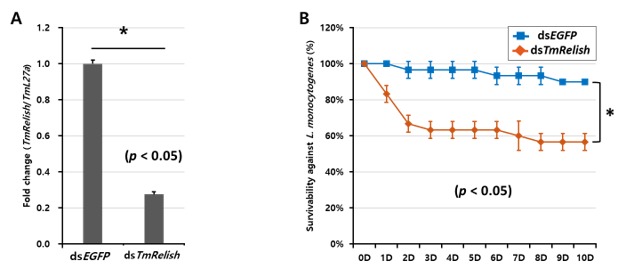
Effect of *TmRelish* silencing on the survival of *T. molitor* larvae after challenging with *L. monocytogenes* for 10 days. Knockdown efficiency of *TmRelish* mRNA, extracted from ds*TmRelish*-injected larvae in comparison ds*EGFP* injected group was measured 3-day post-injection by qRT-PCR (**A**). Survival results for ds*TmRelish*-injected larvae after *L. monocytogenes* challenge are presented as the average of three biological replicates (**B**). The ds*EGFP*-injected groups followed by same microbial infection were used as negative controls. ‘*’ indicates significant differences between ds*TmRelish* and ds*EGFP*-treated groups (*p* < 0.05).

**Figure 3 insects-11-00188-f003:**
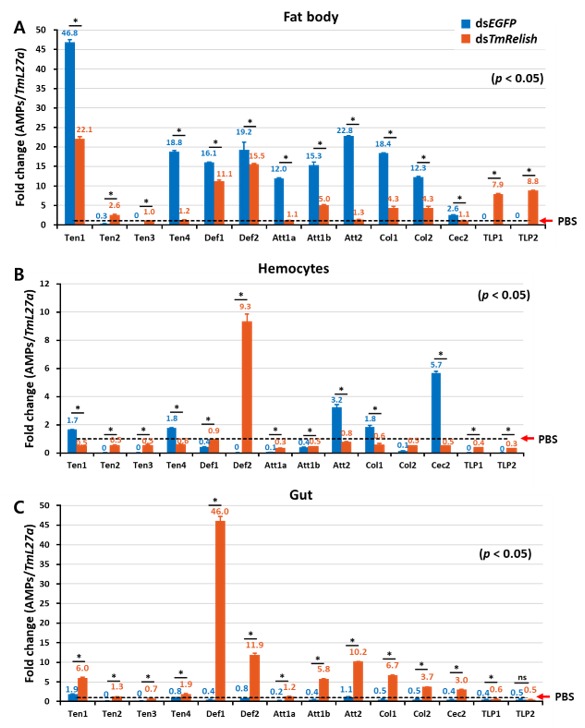
Analysis of AMP genes in the fat body (**A**), hemocytes (**B**), and gut (**C**) of *T. molitor* larvae after *TmRelish* silencing followed by microbial challenge with *L. monocytogenes*. The mRNA levels of 14 AMP genes comprising *TmTenecin-1* (*TmTene1*), *TmTenecin-2* (*TmTene2*), *TmTenecin-3* (*TmTene3*), *TmTenecin-4* (*TmTene4*), *TmDefensin1* (*TmDef1*), *TmDefensin2* (*TmDef2*), *TmColeoptericin-1* (*TmCole1*), *TmColeoptericin-2* (*TmCole2*), *TmAttacin1a* (*TmAtt1a*), *TmAttacin-1b* (*TmAtt1b*), *TmAttacin-2* (*TmAtt2*), *TmCecropin-2* (*TmCec2*), *TmThaumatin-like protein-1* (*TmTLP1*), and *TmThaumatin-like protein-2* (*TmTLP2*) were measured by qRT-PCR. Statistical significance of the fold change in AMP gene expression in *TmRelish*-knockdown larvae compared with negative control and double stranded *EGFP*-treated larvae is indicated by asterisks (*p* < 0.05) and ns = not significant. The number above the bars indicates the AMP transcript level. Error bars indicate the SEM of three biological experiments.

**Figure 4 insects-11-00188-f004:**
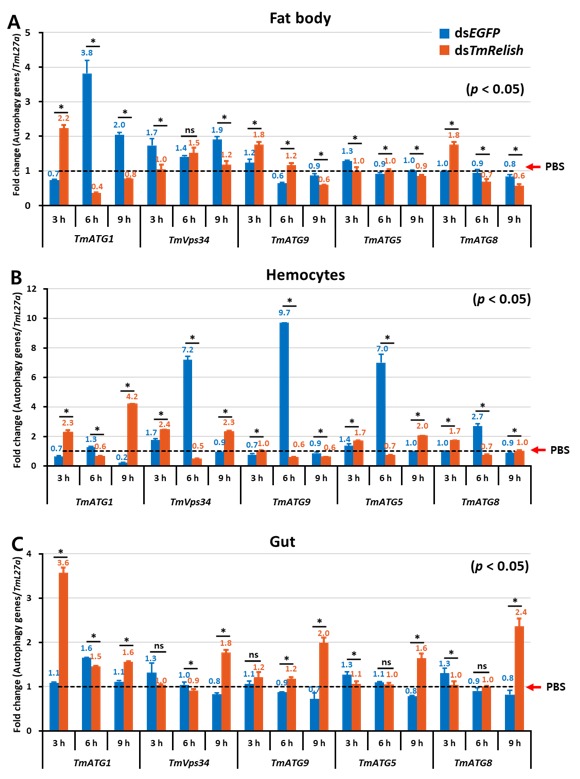
Effect of *TmRelish* gene silencing on the expression level of *T. molitor* autophagy-related genes (*Tm*ATG) in the fat body (**A**), hemocytes (**B**), and gut (**C**) of larvae at 3, 6, and 9 h after *L. monocytogenes* challenge. Transcriptional levels of *TmATG1*, *TmVps34*, *TmATG9*, *TmATG5*, and *TmATG8* were quantified by qRT-PCR. All experiments were performed on three independent sets. asterisks ‘*’ indicate statistically significant differences between the ds*TmRelish*- and ds*EGFP*-treated groups using Student’s t-tests (*p* < 0.05) and ns = not significant.

**Figure 5 insects-11-00188-f005:**
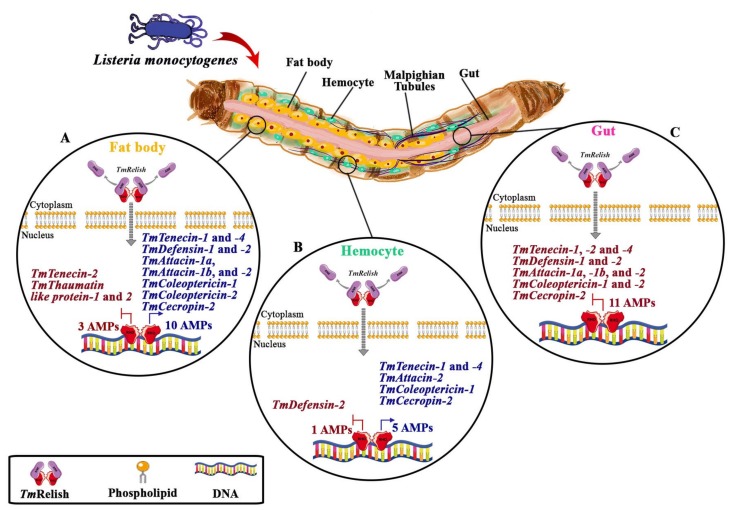
Schematic depiction of *TmRelish*-mediated activation of the immune-deficiency (Imd) pathway in the larval fat body (**A**), hemocytes (**B**), and gut (**C**) by *L. monocytogenes*. Upon intracellular infection, the N-terminus domain of *TmRelish*, the Rel homology domain (RHD), translocates from the cytoplasm to the nucleus to induce the expression of antimicrobial peptides (AMPs) in the larval fat body and hemocytes of *T. molitor*. *Tm*Relish negatively regulates the induction of several AMPs in the fat body, hemocytes, and gut of *T. molitor* larvae in response to *L. monocytogenes*.

**Table 1 insects-11-00188-t001:** Sequences of the primers used in this study.

Primer	Sequence
TmRelish_qPCR_Fw	5′-AGCGTCAAGTTGGAGCAGAT-3′
TmRelish_qPCR_Rv	5′-GTCCGGACCTCATCAAGTGT-3′
TmRelish_Temp_Fw	5′-TGTGGGAAGATTACGGGAAA-3′
TmRelish_Temp_Rv	5′-CAAATTGGCCACGATCTCTT-3′
dsTmRelish_Fw	5′-TAATACGACTCACTATAGGGTGACGTGCACCATCAATA-3′
dsTmRelish_Rv	5′-TAATACGACTCACTATAGGGTGCGTGTTTGGCCTTGAT-3′
dsEGFP_Fw	5′-TAATACGACTCACTATAGGGTACGTAAACGGCCACAAGTTC-3′
dsEGFP_Rv	5′-TAATACGACTCACTATAGGGTTGCTCAGGTAGTGTTGTCG-3′
TmTenecin-1_Fw	5′-CAGCTGAAGAAATCGAACAAGG-3′
TmTenecin-1_Rv	5′-CAGACCCTCTTTCCGTTACAGT-3’
TmTenecin-2_Fw	5′-CAGCAAAACGGAGGATGGTC-3′
TmTenecin-2_Rv	5′-CGTTGAAATCGTGATCTTGTCC-3′
TmTenecin-3_Fw	5′-GATTTGCTTGATTCTGGTGGTC-3’
TmTenecin-3_Rv	5′-CTGATGGCCTCCTAAATGTCC-3′
TmTenecin-4_Fw	5′-GGACATTGAAGATCCAGGAAAG-3′
TmTenecin-4_Rv	5′-CGGTGTTCCTTATGTAGAGCTG-3′
TmDefensin-1_Fw	5′-AAATCGAACAAGGCCAACAC-3′
TmDefencin-1_Rv	5′-GCAAATGCAGACCCTCTTTC-3′
TmDefensin-2_Fw	5′-GGGATGCCTCATGAAGATGTAG-3′
TmDefensin-2_Rv	5′-CCAATGCAAACACATTCGTC-3′
TmColeoptericin-1_Fw	5′-GGACAGAATGGTGGATGGTC-3′
TmColeoptericin-1_Rv	5′-CTCCAACATTCCAGGTAGGC-3’
TmColeoptericin-2_Fw	5′-GGACGGTTCTGATCTTCTTGAT-3′
TmColeoptericin-2_Rv	5′-CAGCTGTTTGTTTGTTCTCGTC-3′
TmAttacin-1a_Fw	5′-GAAACGAAATGGAAGGTGGA-3′
TmAttacin-1a_Rv	5′-TGCTTCGGCAGACAATACAG-3′
TmAttacin-1b_Fw	5′-GAGCTGTGAATGCAGGACAA-3′
TmAttacin-1b_Rv	5′-CCCTCTGATGAAACCTCCAA-3′
TmAttacin-2_Fw	5′-AACTGGGATATTCGCACGTC-3′
TmAttacin-2_Rv	5′-CCCTCCGAAATGTCTGTTGT-3’
TmCecropin-2_Fw	5′-TACTAGCAGCGCCAAAACCT-3′
TmCecropin-2_Rv	5′-CTGGAACATTAGGCGGAGAA-3′
TmThaumatin-like protein-1_Fw	5′-CTCAAAGGACACGCAGGACT-3′
TmThaumatin-like protein-1_Rv	5′-ACTTTGAGCTTCTCGGGACA-3′
TmThaumatin-like protein-2_Fw	5′-CCGTCTGGCTAGGAGTTCTG-3′
TmThaumatin-like protein-2_Rv	5′-ACTCCTCCAGCTCCGTTACA-3′
TmL27a_qPCR_Fw	5′-TCATCCTGAAGGCAAAGCTCCAGT-3′
TmL27a_qPCR_Rv	5′-AGGTTGGTTAGGCAGGCACCTTTA-3′

Note: Underline indicates T7 promoter sequences.

**Table 2 insects-11-00188-t002:** Sequences of the primers used in this study.

Autophagosome Protein Complex	Autophagy-Related Genes	Sequence
Initiation	TmATG1-qPCR-FwTmATG1-qPCR-Rv	5′-TTGGCCGATTATCTCAACGC-3′5′-TTCATGGCGCCAGCTAATTG-3′
Nucleation	TmVps34-qPCR-FwTmVps34-qPCR-Rv	5′-AGCACCAAGGAGTTCCAGGAA-3′5′-ATGTTGCCGTTGTGTCTGTC-3′
Recycling	TmATG9-qPCR-FwTmATG9-qPCR-Rv	5′-AGTGCGAAAACGGCAAACTG-3′5′-ATGCTGCTCTGATTCTGCAC-3′
Elongation	TmATG5-qPCR-FwTmATG5-qPCR-Rv	5′-GGGCTGTGAATCGAAAGTTG-3′5′-GTTTTGCGGTGTCCATCTTC-3′
Completion and extension	TmATG8-qPCR-FwTmATG8-qPCR-Rv	5′-AAGATCCGCCGAAAGTATCC-3′5′-AACTGGCCGACTGTCAAATC-3′

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
