# Peer review of "Two Roles for the Tenebrio molitor Relish in the Regulation of Antimicrobial Peptides and Autophagy-Related Genes in Response to Listeria monocytogenes"

_insects, 2020, doi:10.3390/insects11030188_

Round 1
Reviewer 1 Report
1. Brief summary:
- Using RNAi, the authors show that NF-κB transcription factor Relish, which is a part of the IMD pathway, is responsible for both AMP gene regulation and autophagy responses in fat body and hemocytes in molitor larvae. Relish’s role in ATG gene regulation in T. molitor was previously known, but the authors show that it also plays a dual-role by regulating AMP expression in response to Listeria infection.
2. Broad comments:
- The abstract and introduction are clearly written and establish the background and knowledge gap the authors seek to fill. The methods are standard bacteria culturing, qRT-PCR, RNAi, etc. Proper RNAi controls were utilized and experiments were carried out in triplicate. Figures are easy to follow and well-presented, although figures 3-5 and 6-8 would be better presented in condensed form as two figures with 3 panels each since expression of the same panel of AMPs was examined in all three tissues. The presentation of the gene expression data throughout the manuscript focuses too heavily on instances where relish silencing downregulates gene expression relative to controls, as is seen with 10/14 AMP genes in the fat body. However, it is not clear why this focus is justified, given that relish silencing often had the opposite effect, leading to an upregulation in AMP and ATG genes, especially in the gut. These sections should be re-framed to better capture this diversity of responses and relish’s role as both a positive and a negative regulator of gene expression in all tissues (particularly in the gut for the latter). The same applies to the summary of the expression results in Figure 9, which need to be expanded. Furthermore, the authors should discuss other instances where relish has been shown to negatively regulate gene expression (such as was found here recently: PMID: 26847920), as this is typically seen as the role of Caspar in the Imd pathway and would thus be an important and interesting finding. Also, the authors should take care in what is claimed throughout the manuscript for what appear to be relatively negligible fold changes in gene expression, keeping in mind the distinction between mere statistical, and biological significance.
3. Specific comments:
- Lines 2-5: TmRelish is presented as only a positive regulator of AMP and ATG genes in the title, but the data shows that it is also a negative regulator of gene expression, particularly in the gut. The title should be broadened to encompass the range of this study’s findings.
- Line 16: Change “opaque” to “unclear.”
- Line 33: Consider removing “intensively” or substitute with a more suitable adjective.
- Line 58: Remove “are.”
- Lines 58-62. Consider breaking these into multiple sentences.
- Line 90: Can the authors provide a brief comment as to why molitor was maintained in the dark, as opposed to in a regular day night cycle?
- Lines 106-108: How, precisely, were the larvae challenged? Injection? Insect pin prick? Feeding?
- Lines 108-110: Please describe in greater detail the methods used for dissecting the fat body, hemocytes, and the gut, being careful to mention how the fat body and hemocyte tissues were isolated so as to avoid cross contamination.
- Line 159: “Larval” does not need to be capitalized.
- Line: 198: Meaning of the phrase “Face to L. monocytogenes challenge” is unclear. Consider revising.
- Figure 1A is missing a “PBS” label for the black dashed line, as in panels B and C.
- Line 243: I do not think this needs to be in bold, but perhaps the word “increase” could be italicized.
- Line 278: Remove comma after “infection.”
- Lines 301-302: The “marked decrease in TmATG9” expression represents a fold-change difference of only 0.3 (0.9 compared to 0.6). While this difference is listed as statistically significant, so are all of the other differences, including even the 9 h TmATG8 groups, for which there was only a very small 0.1 fold change (0.9 to 1.0) in expression. Furthermore, both of these fold changes for these genes at 9 h are below even PBS-treated larvae expression. Thus, these changes do not appear to be statistically, much less biologically significant, and thus describing these as “marked” differences does not appear to be justified. Similar issues occur in Figs. 3-6 and 8.
- Lines 316-318: These lines appear to be from the author guidelines description of the results section and should be removed.
- In Figure 8, as with the other figures, the authors tend to highlight only instances in which dsTMRelish downregulates gene expression relative to the dsEGFP However, there are many instances of upregulation in the dsTMRelish groups in all of the tissues, especially in the gut, and one wonders why these are not highlighted equally. One could argue that this is as prominent a phenotype as the downregulation upon relish silencing.
- Line 346: Consider using “Congruent with our finding” or a similar phrase as opposed to “In congruence with our finding.”
- Lines 370-373: Please discuss similar instances in the fat body and hemocytes.
- Line 377: Increased expression of TmTene3 and TmDef2 upon silencing of relish is mentioned, but not explained. Please attempt to address this trend both here and elsewhere in the manuscript.
- Lines 393-394: Again, caution should be used in claiming that TmATG8 was downregulated at 6 g pi in the fat body, given how minor the change was (0.9 to 0.7, so only a 0.2 fold change).
- Line 399: The previous comment also applies here for the 9 h pi TmATG8 fat body expression data.
- Line 412: Replace “Although” with “However.”
- Figure 9 is beautifully rendered, but TmRelish’s role as a positive regulator of AMP/ATG genes in the fat body and hemocytes alone are highlighted. This figure should be expanded to encompass the expression findings in the gut as well as all instances of negative regulation in each of these tissues so that readers can get a full picture of the results of your expression analyses.
Author Response
Dear Reviewer 1,
Please see the attachment.
Best regards,

Reviewer 2 Report
In this manuscript, the authors showed that Tenebirio molitor Relish (TmRelish) were involved in regulation of antimicrobial peptide (AMP) genes and autophagy-related (ATG) genes in response to invasion of Listeria monocytogenes.
Firstly the author investigated mRNA amounts of TmRelish after L. monocytogenes invasion. Consequently, the author showed TmRelish knockdown larvae were more susceptible to L. monocytogenes infections than control larvae.
The authors measured mRNA amounts of 14 AMP genes and 5 ATG genes in three tissues (fat body, hemocytes and guts) from TmRelish knockdown and control knockdown larvae. By comparing the amounts of total 19 genes between the TmRelish knockdown and control knockdown samples, In fat body 10 AMP gene were induced by L. monocytogenes infection, in which TmRelish can be involved. In hemocytes, the same cases were observed in some AMP genes. On the other hand, TmRelish knockdown cause increases of the levels of several AMP genes in gut, suggesting that TmRelish down-regulated some AMP genes in gut. In case of ATG genes, TmRelish knockdown brought about changes induction levels of some ATG genes in the three tissues. Some ATG genes induction by was surpressed in the three tissues of several time points by TmRelish while level of other ATG genes was increased. Taken together these results, the authors concluded TmRelish contribute to immune and autophagic reactions against L. monocytogenes.
In my opinion, these findings are important for insect immunology communities, and the manuscript including the findings is worth to publishing in "insects" journal. I suggest several revisions including addition of several data to improve the manuscripts before publication.
- In table 1 and 2, the authors provide sequences of primer pairs for synthesizing dsRNA or qRT-PCR. However the sequences or IDs in public database of sequences utilized for designing these primers are not shown. For reproducibility, the author must show them.
- The author measured amounts of AMP genes in a single time point (Figs. 3, 4 and 5), at 24 post infection. Previous reports about insect immunology showed that profiles of mRNA inductions by several microbes varies between AMP genes. So I think that amounts of 14 AMP genes between TmRelish knockdown and control larvae at different time point must be added.
- The author showed that TmRelish contributed to autophagic reaction against L. monocytogenes. In my opinion, additions of several morphologic data related to autophagy must be preferred for the conclusion.
Author Response
Dear Reviewer 2,
Please see the attachment.
Best regards,

Round 2
Reviewer 2 Report
Author addressed properly reviewer's comments.
I recommend this manuscript should be published.